# Immune subtyping of lymph node metastasis-negative colorectal cancer reveals biomarkers for prognosis and immunotherapy response

Wenliang Yuan⬤*, Li Liu

College of Data Science, Jiaxing University, Jiaxing, Zhejiang, China

* yuanwl@zjxu.edu.cn

## Abstract

### Background

Lymph node metastasis (LNM) is a key prognostic factor in colorectal cancer (CRC), and early lymph node status assessment is crucial for prognosis and immunotherapy decisions. However, the immune characteristics of LNM-negative CRC remain poorly understood.

### Methods

Using machine learning algorithms, we identified and analyzed two immune subtypes (C1 and C2) in 244 LNM-negative CRC samples, with validation in 458 additional samples, and evaluated their immune characteristics and functional pathways.

### Results

Subtype C1 exhibited high immune scores and responsiveness to immune check-point blockade, while subtype C2 showed a favorable prognosis and increased immune cell infiltration, indicating it represents an earlier CRC stage. Experimental validation revealed that *PPFIA4* knockdown in C1 significantly suppressed CRC cell proliferation and migration.

### Conclusions

These findings provide insights into personalized immunotherapy strategies for early-stage CRC patients and have potential clinical application value.

## Introduction

Colorectal cancer (CRC) is the third most prevalent malignancy globally, comprising approximately 10.2% of cancer cases and causing a 9.2% mortality rate [1]. Metastasis occurs in 50–60% of CRC patients, frequently resulting in poor surgical outcomes

**Data availability statement:** All relevant data are within the paper and its Supporting information files.

**Funding:** This work was funded by The Jiaxing Science and Technology Plan Project, 2023AY11057 awarded to Dr. Wenliang Yuan, The Jiaxing Science and Technology Plan Project, 2024AY10048, awarded to Dr. Li Liu, and The General Research Project of the Zhejiang Provincial Department of Education, Y202352288, awarded to Dr. Wenliang Yuan.

**Competing interests:** The authors have declared that no competing interests exist.

and diminished survival due to disease recurrence [2,3]. Translymphatic spread is a common metastatic pathway, significantly impacting long-term survival [4]. Therefore, early evaluation of lymph node status is crucial for accurate prognosis and treatment planning. However, the TNM (Tumor-Node-Metastasis) staging system, a primary prognostic tool for CRC, often lacks granularity, as patients within the same stage can exhibit substantial variability in survival outcomes post-surgery [5].

Lymph node metastasis-negative (LNM-negative) CRC constitutes a substantial subset of early-stage cases and generally confers a favorable prognosis. However, notable clinical heterogeneity persists, with recurrence occurring in some patients. Emerging evidence suggests that LNM-negative tumors possess distinct immune features, yet systematic immune subtyping remains lacking [6]. Immunotherapy, especially PD-1/PD-L1 blockade, has become a pivotal strategy in cancer treatment by enhancing anti-tumor immune responses [7]. Immune cells play a vital role in CRC progression, influencing both tumor initiation and therapeutic efficacy [8]. The tumor microenvironment (TME), especially interactions between immune and CRC cells, shapes tumor behavior and modulates treatment response [9,10]. For example, M2-polarized macrophages contribute to chemotherapy resistance and promote CRC cell migration [11]. However, immunotherapy is argely limited to tumors with specific features, such as defective mismatch repair (dMMR) or high microsatellite instability (MSI) [12]. While recent classifications of MSI-high CRC have identified distinct immune subtypes [13], the immunological landscape of LNM-negative CRC remains largely unexplored. A comprehensive classification system for LNM-negative CRC could facilitate the identification of biomarkers for personalized immunotherapy.

In this study, we utilize machine learning algorithms to identify novel immune subtypes in LNM-negative CRC using large-scale datasets. These subtypes are characterized by their prognostic features, immune profiles, transcriptomic patterns, clinical traits, immune cell infiltration, and genetic mutations. We also introduce an immune checkpoint inhibition (ICI) score to assess immune states and investigate the functional role of *PPFIA4*, which encodes the protein liprin-α4, in CRC cell proliferation and migration. Our findings provide novel insights into the refined classification of CRC immune subtypes and highlight potential biomarkers for optimizing immunotherapy, particularly in LNM-negative patients.

## Materials and methods

### Data sources and preprocessing

Colorectal cancer (CRC) data were sourced from The Cancer Genome Atlas (TCGA; 244 samples) and Gene Expression Omnibus (GEO; 302 from GSE39582 and 156 from GSE103479). These datasets are publicly available and anonymized, meaning that no direct human subjects or animal studies were involved. As such, no ethics approval or patient consent is required for this study. After excluding genes with low median absolute deviation (MAD), a univariate Cox proportional hazards model was applied to link gene expression with overall survival. Focusing on 2,995 immune-related genes [14], we employed non-negative matrix factorization (NMF) for

clustering, as it is well-suited for high-dimensional transcriptomic data and has been widely used to identify biologically interpretable cancer subtypes. The clustering results were validated using t-distributed stochastic neighbor embedding (t-SNE). Genes positively associated with clusters were classified as ICI feature A, while the remaining were designated as feature B. The Boruta algorithm [15] further refined these features, and principal component analysis (PCA) was applied to derive the first principal component from each set, summarizing key dimensions into interpretable components to construct the ICI score (ICI score $= PC1_A - PC1_B$).

## Analysis of biological functions and immune landscape of CRC subtypes

Gene set variation analysis (GSVA) was used to evaluate sample-pathway relationships by calculating enrichment scores with 29 CRC-related gene signatures across four categories: Signatures, Canonical, Immune, and Metabolism [16] Gene Set Enrichment Analysis (GSEA) was then performed to assess statistical significance between high and low ICI score groups [17]. To characterize CRC subclasses, immune infiltration was estimated using both the MCP-counter and single-sample GSEA (ssGSEA), which calculated enrichment scores for 13 immune cell groups [18]. Differentially expressed mRNAs between LNM-negative CRC subclasses were identified using edgeR with a criterion of $|log_2$ fold change$| >= 1.5$ and FDR $< 0.01$ [19]. Immune and stromal scores were computed using the ESTIMATE algorithm [20].

## Cell culture, transfection, and qPCR analysis

Human colorectal cancer cell lines SW480 (FH0022) and HCT-116 (FH0027) were obtained from Fuheng Biotechnology (Shanghai, China). SW480 cells were cultured in L15 medium, and HCT-116 in McCoy's 5A, both supplemented with 10% FBS and 1% penicillin/streptomycin. Cells were seeded in 6-well plates and transfected at 70−80% confluence with Lipofectamine 3000 using si-NC or si-*PPFIA4*. After 48 hours, RNA was extracted for qPCR analysis. RNA was reverse-transcribed and analyzed by qPCR on an X960 Real-time Thermal Cycler using SYBR Green, with GAPDH as the reference gene. Gene expression was quantified by the $2-\Delta\Delta Cq$ method, using primers for *PPFIA4* (F: 5'-CTCTG CGGATGTTGTCTCCC-3', R: 5'-ATGCTGCCACTGGTTACACG-3') and *GAPDH* (F: 5'-GGAGCGAGATCCCTCC AAAAT-3', R: 5'-GGCTGTTGTCATACTTCTCATGG-3').

## Protein analysis, cell proliferation, and migration evaluation

Proteins were extracted and quantified using a BCA assay, separated by SDS-PAGE, and transferred to PVDF membranes. After blocking, membranes were incubated with primary antibodies against liprin-α4 and GAPDH, followed by HRP-conjugated secondary antibodies. Signals were detected using an enhanced chemiluminescence (ECL) kit. Cell proliferation was assessed using an MTT assay on days 1, 3, 5, and 7, with optical density measured at 570 nm. A significant reduction in proliferation was observed in SW480 cells on days 5 and 7 ($P < 0.05$) and in HCT-116 cells from day 3 ($P < 0.01$). For the wound healing assay, SW480 and HCT-116 cells were grown to full confluence in 6-well plates. A sterile 200 μL pipette tip was used to manually create a linear scratch through the cell monolayer. After washing with PBS to remove detached cells, the remaining cells were cultured in serum-free medium supplemented with mitomycin C (10 μg/mL) to inhibit proliferation. Images were captured at 0, 24, and 48 hours using an inverted microscope, and the wound area was quantified using ImageJ software. All in vitro experiments were performed in at least three independent biological replicates. Technical replicates were included where applicable. MTT assays, wound healing assays, qPCR, and Western blot experiments were repeated to ensure robustness and reproducibility.

## Statistical analyses

All statistical analyses and computational workflows were reviewed in consultation with a professional biostatistician to ensure methodological rigor and analytical robustness. Analyses were conducted using R (version 3.5.0), with chi-square

tests applied to compare immune scores, clinical data, and mutation rates. A univariate Cox proportional hazards model was used to identify survival-associated genes, and Kaplan–Meier survival curves were generated, with P < 0.05 considered statistically significant.

## Results

### Identification and validation of two molecular subtypes in LNM-negative CRC

We selected 109 candidate genes from immune-related and CRC survival-associated genes for non-negative matrix factorization (NMF) analysis (S1 Table). NMF consensus clustering of the TCGA dataset identified two distinct molecular subtypes ($k=2$) with clear boundaries in the consensus matrix (Fig 1A). To ensure comparability across datasets and minimize potential batch effects, we performed standardized data preprocessing before conducting cross-cohort validation. Chi-square filtering revealed 80 genes significantly associated with the clustering results (S1 Table). Independent analyses of GEO datasets (GSE39582 and GSE103479) confirmed two molecular subclasses in LNM-negative CRC (Fig 1D, S1A Fig.). T-SNE analysis further supported the consistency of the clustering across both datasets (Figs 1B and 1E).

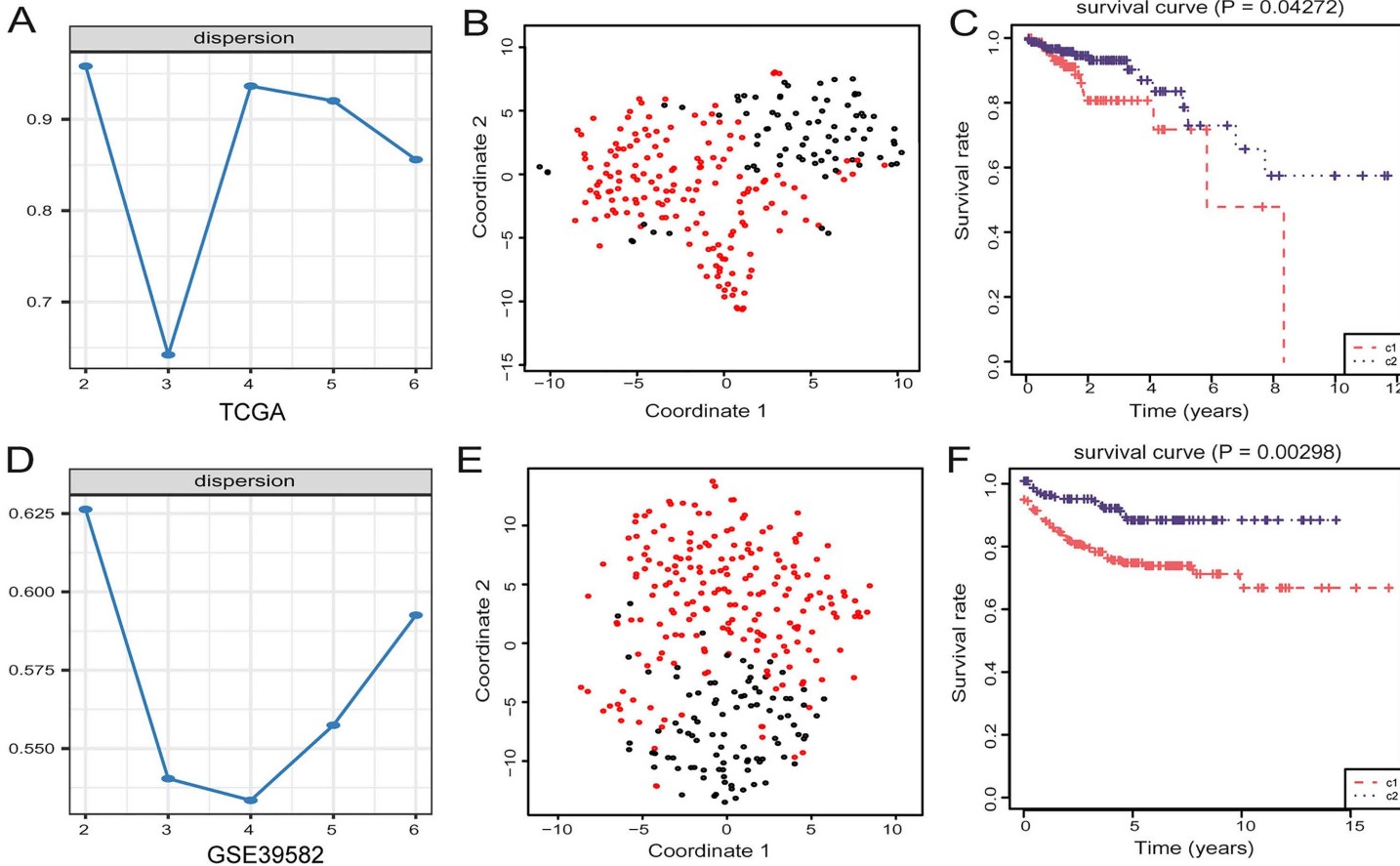

**Fig 1. Use non-negative matrix factorization (NMF) to identify subclasses in LNM negative CRC. (A)** Use immune-related genes to perform NMF clustering on the TCGA. Cophenetic correlation coefficient for k = 2-6 is shown. **(B)** T-SNE analysis on the TCGA supports dividing it into two subclasses. **(C)** Overall survival (OS) analysis of the two subclasses (C1 and C2) in TCGA. **(D)** Use immune-related genes to perform NMF clustering on the GSE39582. **(E)** T-SNE analysis on the GSE39582 supports dividing it into two subclasses. **(F)** OS analysis of the two subclasses (C1 and C2) in GSE39582, with statistical significance assessed by log-rank test.

Prognostic differences were observed in both TCGA and GEO datasets (Figs 1C and 1F), with significant differences in "Pathologic stage" and "Pathologic M" proportions between subclasses (Table 1), and in "Pathologic T" and "Vital status" in GSE39582. Together, these results demonstrate that the identified molecular subtypes are robust, reproducible across independent datasets, and are associated with distinct gene expression profiles and prognostic outcomes in LNM-negative CRC.

## Correlation of the LNM-negative CRC subtypes with immune-associated signatures

To characterize the immune features of the two LNM-negative CRC subtypes, we performed differential gene expression analysis and identified 160 subtype-specific signature genes (S2 Table). Gene ontology enrichment revealed significant involvement of these genes in immune-related processes, including T cell and lymphocyte proliferation (S3 Table), and pathways such as Th17 cell differentiation (S4 Table). GSVA analysis of 29 CRC-relevant immune signatures revealed significant differences in canonical pathways like MAPK, as well as immune pathways such as immune response, PD1 activation, and complement activation in C2 (Fig 2A, S5 Table). Scoring analysis showed that C2 had lower immune scores (reflecting immune cell infiltration), stromal scores (indicating stromal content in the tumor microenvironment), and cytolytic scores (quantifying cytotoxic T cell activity), with no significant difference in proliferation scores (measuring tumor cell division rates), compared to C1 (Figs 2B–2E). These findings suggest distinct immune activity profiles between the subtypes.

**Table 1. Clinical Characteristics of patients with distinct classification in TCGA and GSE39582.**

| | TCGA | | | GSE39582 | | |
|---|---|---|---|---|---|---|
| | C1 | C2 | P | C1 | C2 | P |
| **LNM-negative** | 81 | 163 | | 207 | 95 | |
| **Age(years)(%)** | | | 0.071 | | | 0.265 |
| > 68 | 36(44.4) | 82(50.3) | | 92(44.4) | 49(51.6) | |
| <= 68 | 45(55.6) | 61(49.7) | | 115(55.6) | 46(48.4) | |
| **Gender(%)** | | | 0.494 | | | 0.803 |
| Male | 43(53.1) | 95(58.3) | | 116(56.0) | 55(57.9) | |
| Female | 38(46.9) | 68(41.7) | | 91(44.0) | 40(42.1) | |
| **Pathologic stage(%)** | | | **0.027** | | | **0.019** |
| Stage I | 17(21.0) | 56(34.4) | | 21(10.1) | 16(16.8) | |
| Stage II | 61(75.3) | 95(58.3) | | 174(84.1) | 79(83.2) | |
| Stage III | 0(0.0) | 0(0.0) | | 0(0.0) | 0(0.0) | |
| Stage IV | 1(1.2) | 7(4.3) | | 12(5.8) | 0(0.0) | |
| **Pathologic T(%)** | | | 0.074 | | | **0.039** |
| T1-T2 | 18(22.2) | 57(35.1) | | 20(9.6) | 18(18.9) | |
| T3-T4 | 63(77.8) | 106(64.9) | | 186(90.4) | 77(81.1) | |
| **Pathologic M(%)** | | | **0.001** | | | **0.016** |
| M0 | 71(87.7) | 141(86.5) | | 194(93.7) | 94(98.9) | |
| M1 | 1(1.2) | 7(4.3) | | 13(6.3) | 0(0.0) | |
| MX | 9(11.1) | 15(9.2) | | 0(0.0) | 1(1.1) | |
| **Vital status(%)** | | | 0.136 | | | **0.001** |
| Alive | 68(84.0) | 148(91.8) | | 152(73.4) | 85(89.4) | |
| Dead | 13(16.0) | 15(9.2) | | 55(26.6) | 10(10.6) | |

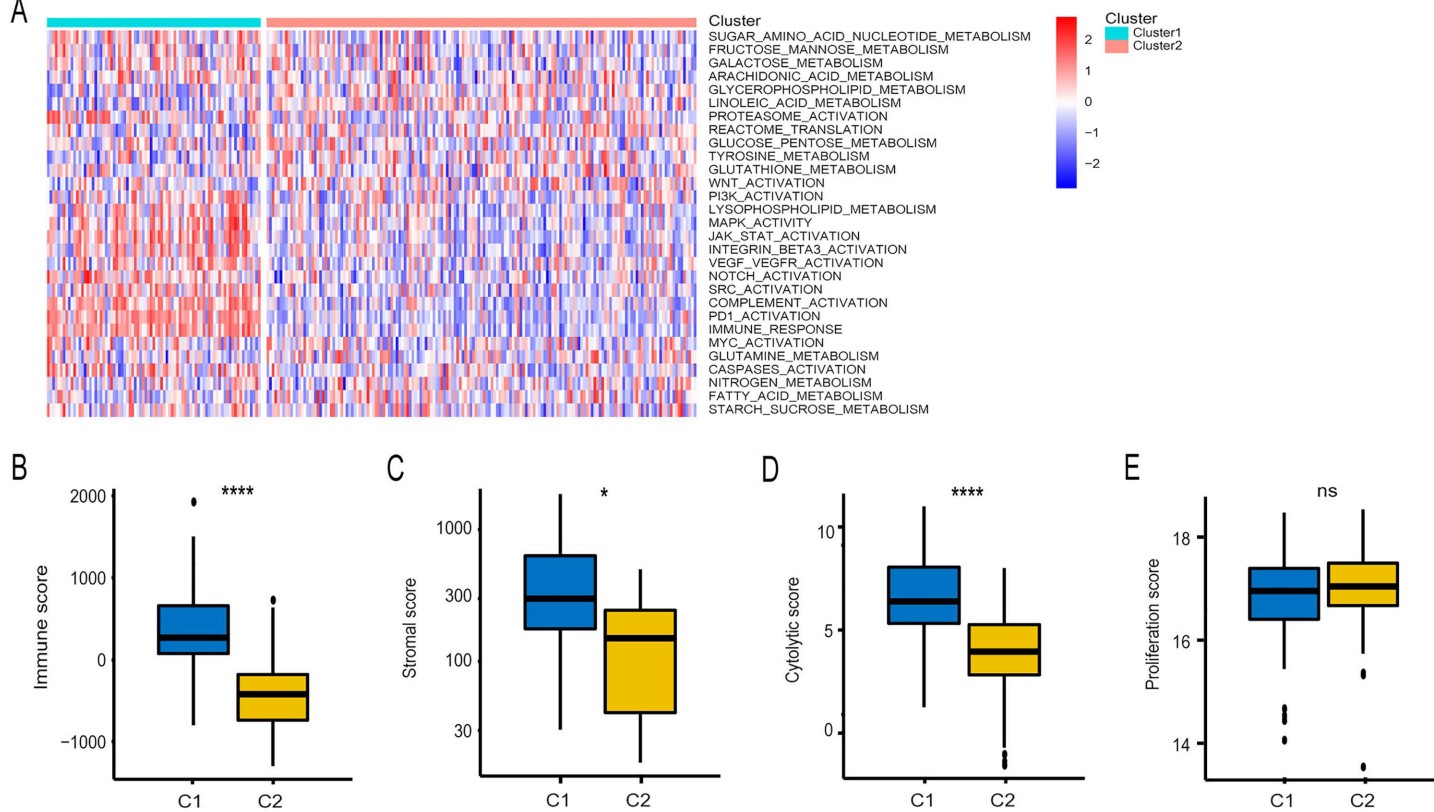

**Fig 2. Association between immune-associated signatures and LNM-negative CRC subclasses. (A)** Heatmap of specific immune-associated signatures. Boxplots for immune score **(B)**, stromal score **(C)**, cytolytic score **(D)**, and proliferation score (E) in LNM-negative CRC subclasses (C1 and C2). Statistical differences were assessed using the Kruskal–Wallis test; asterisks indicate significance levels (ns = not significant, * P < 0.05, **** P < 0.0001).

### Immune cell infiltration and immune checkpoint profiles in LNM-negative CRC Subtypes

Given the observed immune score differences, we further evaluated immune cell infiltration using MCP counter and ssGSEA, calculating the abundance of 13 immune cell types in each subtype (Fig 3A). Consistent with C1's stromal enrichment, 12 immune cell types (excluding neutrophils) were more abundant in C1, including T cells, CD8+ T cells, cytotoxic lymphocytes, and various myeloid cells (Fig 3B, S6 Table). To investigate the immune checkpoint landscape, we examined the expression of 19 clinically targetable immune checkpoint genes. C1 showed higher expression of 17 checkpoint genes compared to C2, with TNFSF14 and TNFSF15 as exceptions(Fig 3C, S7 Table). Furthermore, TIDE algorithm predictions indicated higher T cell exhaustion in C1, implying that patients in this subclass may benefit more from immune checkpoint inhibition therapy (Fig 3D, S8 Table).

### Genomic mutations and immune response differences between LNM-negative CRC subtypes

Somatic mutations are a hallmark of malignancy. We identified significant mutation differences between LNM-negative CRC subtypes (Fig 4A). Both subtypes showed high mutation frequencies in genes like APC, TTN, and TP53, but with differing rates. *KRAS* mutations were detected in 67% of C2 samples but only 39% of C1, and were absent from C1's top 10 mutated genes. Similarly, *BRAF* mutations were more frequent in C2 (S9 Table), indicating distinct genomic profiles. TMB, associated with a stronger immune response, was higher in C1 (Fig 4B). In addition, we also found that C1 had a

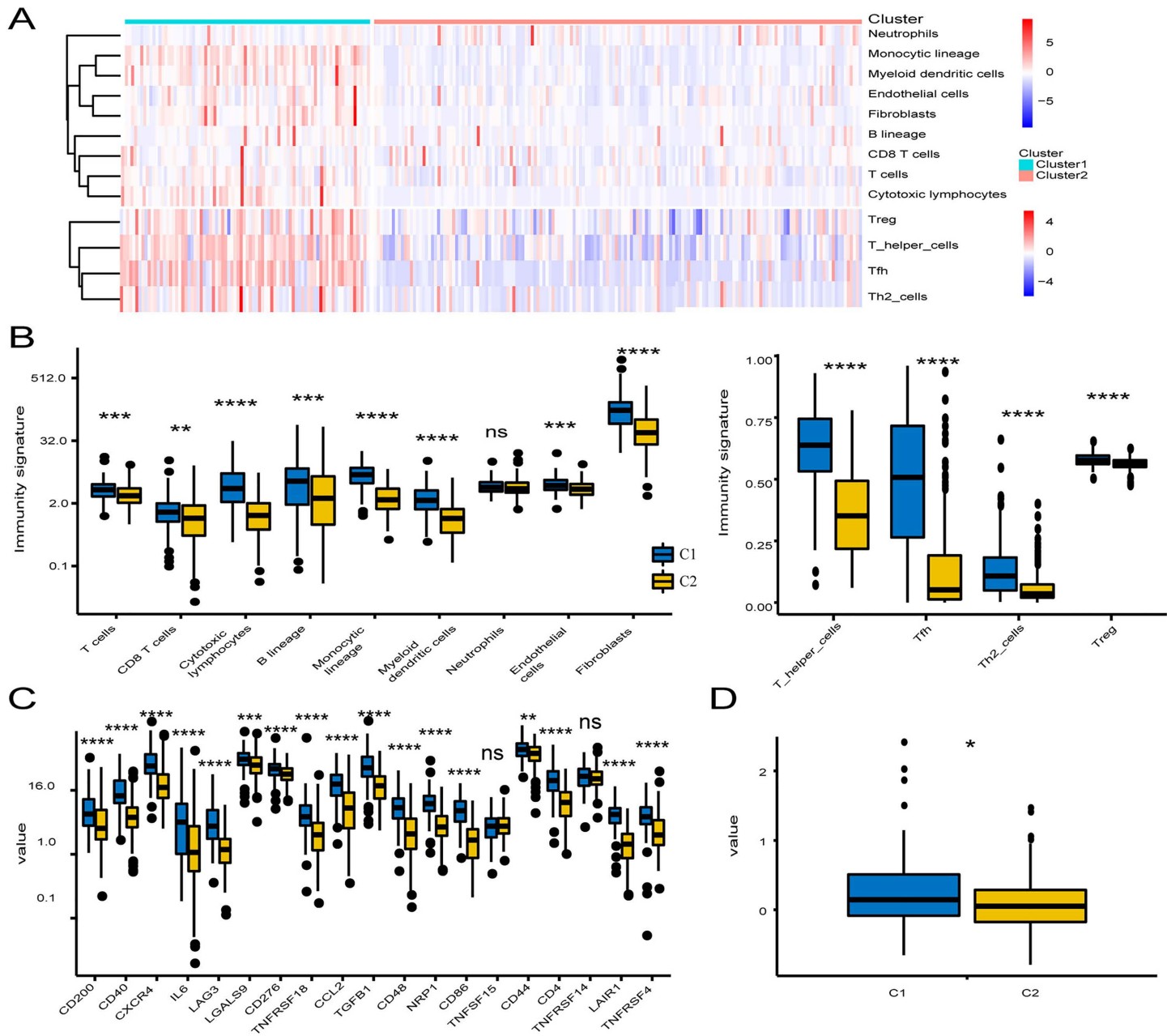

**Fig 3. Correlation between LNM-negative CRC subclasses and immune infiltration. (A)** Heatmap of immune and stromal cell populations. **(B)** Boxplot of immune and stromal cell population abundance in subclasses C1 and C2. **(C)** Expression levels of 19 immune checkpoint genes across subclasses. **(D)** Boxplot of TIDE scores for the two subclasses. Statistical differences were assessed using the Kruskal–Wallis test; asterisks indicate significance levels (ns = not significant, * P < 0.05, ** P < 0.01, **** P < 0.0001).

greater neoantigen burden than C2 (Fig 4D), with C2 showing significantly lower neoantigen and neoantigen origin protein burdens (Fig 4C). CNV analysis confirmed these differences (Fig 4E), suggesting that C1's higher mutation load increases neoantigen production, activating more T cells and enhancing immune response.

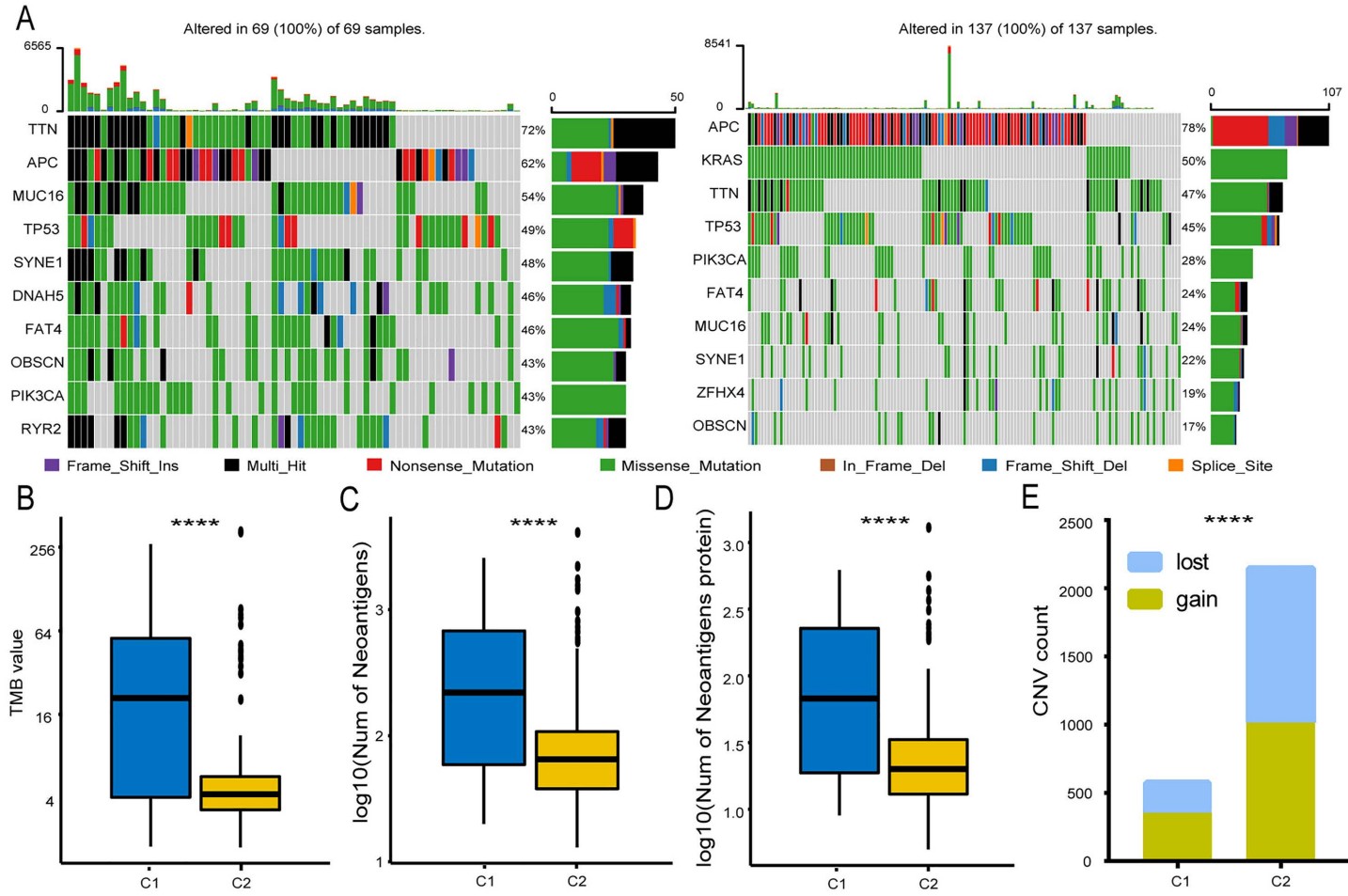

**Fig 4. Association between LNM-negative CRC subclasses and mutations, TMB, neoantigens, and copy number variation (CNV). (A)** Top 10 most mutated genes in subclasses C1 (left) and C2 (right). Boxplots for tumor mutational burden (TMB) **(B)**, neoantigen peptides **(C)**, and neoantigen-related proteins (D) in LNM-negative CRC subclasses. **(E)** CNV distribution in C1 and C2. Statistical differences were assessed using the Kruskal–Wallis test, with asterisks indicating significance levels (**** P < 0.0001).

## Prognostic value of ICI scores and pathway enrichment in LNM-negative CRC

To quantify the immune contexture of LNM-negative CRC, we constructed ICI scores by correlating SGRCs with molecular sub-types in the TCGA and GSE39582 cohorts. ICI scores were computed based on the expression of signature gene sets A and B. Significant differences in ICI scores were observed between the two subtypes(Figs. 5A-5B, S10 Table). Using the optimal cut-off, patients were divided into high and low ICI score groups. Kaplan-Meier survival analysis showed better overall survival for high ICI score patients (Figs 5C-5D). GSEA revealed limited pathway enrichment in the high ICI score group, with only 4 pathways in TCGA (Fig 5E) and the RIBOSOME pathway in GSE39582 (Fig 5F). In contrast, the low ICI score group showed enrichment in multiple immune-related pathways, such as B cell receptor signaling and intestinal immune network for IgA production.

## Expression and functional analysis of *PPFIA4* in CRC subtypes

To identify subtype-specific biomarkers, we performed differential expression analysis between the C1 and C2 subtypes using TCGA and GSE39582 datasets. As shown in S2 Table, PPFIA4 was among the most significantly upregulated

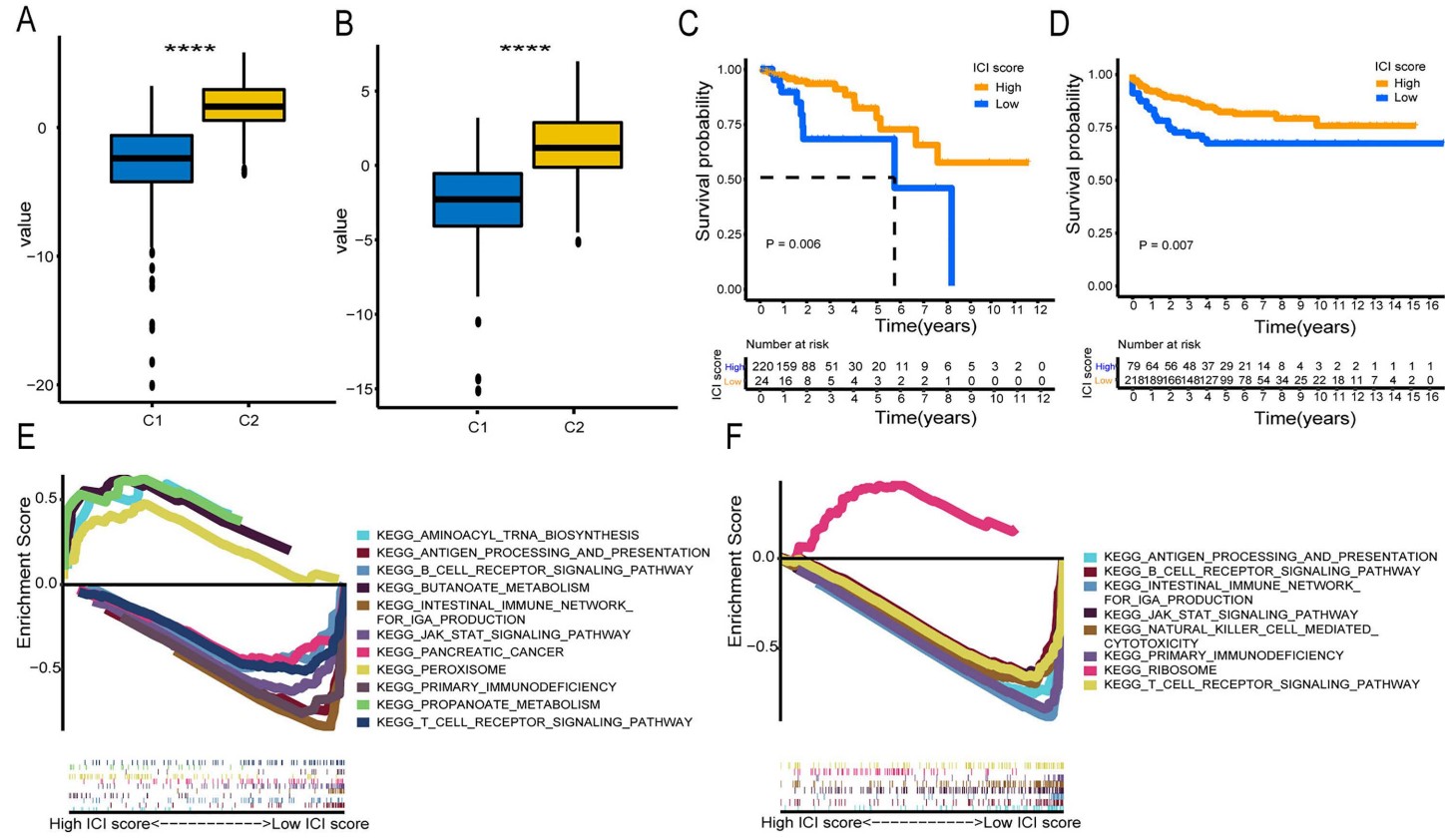

**Fig 5. Construction of the ICI score. (A, B)** Boxplots of ICI scores for the two subcategories in TCGA and GSE39582 datasets. **(C, D)** Survival analysis for high and low ICI subgroups in TCGA and GSE39582. **(E, F)** Enrichment plots for high and low ICI groups in TCGA and GSE39582, with the upper and lower halves of the horizontal axis representing the low and high ICI score subgroups, respectively. Statistical differences were assessed using the Kruskal–Wallis test, with asterisks indicating significance levels (**** P<0.0001).

genes in subtype C1 compared to C2. PPFIA4, which encodes the protein liprin-α4, has been implicated in cancer-related processes such as glycolysis and immune regulation [21,22]. Given its subtype-specific expression pattern and potential biological relevance, we further investigated its functional role in CRC cells through in vitro experiments.PPFIA4 inhibition via siRNA transfection led to a marked reduction in its mRNA (Fig 6A) and protein levels (Fig 6B). Proliferation (MTT) and wound healing assays showed that PPFIA4 inhibition significantly reduced cell proliferation and migration (Fig 6C, 6D). Collectively, these findings indicate that PPFIA4 plays a key role in colorectal cancer cell proliferation and migration, making it a potential therapeutic target.

## Discussion

Lymph node metastasis (LNM) is a critical prognostic factor in CRC, influencing clinical treatment strategies, particularly in rectal cancer. Despite advances, the 5-year survival rate post-surgery remains suboptimal [23]. Recent studies emphasize the potential of immunotherapy, with PD-L1/PD-1 interactions modulating T cell exhaustion [24]. In this study, we identified two distinct LNM-negative CRC subtypes (C1 and C2) based on 2,995 immune-related genes and machine learning, and demonstrated their prognostic, immune, and mutational differences. To further stratify patients, we constructed an ICI score, serving as a prognostic biomarker and a predictor of immunotherapy response. Notably, higher ICI scores were associated with lower immune and stromal scores, whereas lower ICI scores correlated with increased immune infiltration,

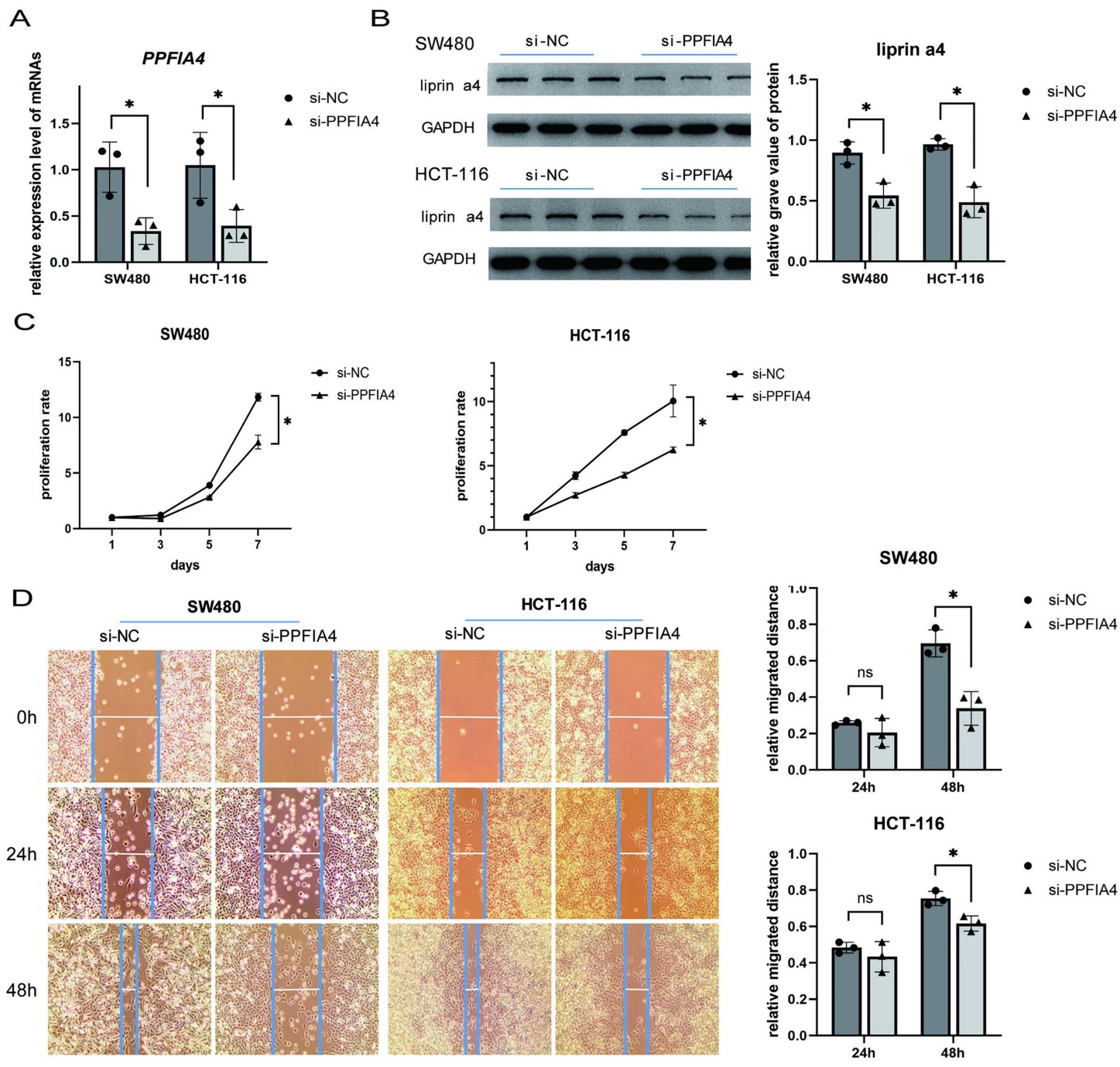

**Fig 6. PPFIA4 Knockdown Suppresses Expression, Proliferation, and Migration in Colorectal Cancer Cells.** (A) qPCR analysis showing a significant reduction in PPFIA4 mRNA expression in the si-PPFIA4 group.(B) Western blot (WB) validation of decreased liprin α4 protein expression in the si-PPFIA4 group, confirming transfection efficiency.(C) MTT assay demonstrating a marked reduction in cell proliferation upon PPFIA4 knockdown.(D) Wound healing assay showing impaired migration. Scratches were created using a sterile 200 μL pipette tip, and mitomycin C (10 μg/mL) was added to inhibit proliferation. Images were taken at 0, 24, and 48 hours. Wound area was quantified using ImageJ. (* $P < 0.05$, ** $P < 0.01$,**** $P < 0.0001$).

suggesting the utility of our subtype classification in guiding immunotherapy strategies. Subtype C1 exhibited high immune scores and responsiveness to immune checkpoint inhibitors, while C2, with favorable prognosis and high ICI scores, may represent an earlier stage of CRC. This immune-based subtyping approach differs from broader classifications like the Consensus Molecular Subtypes (CMS) by specifically targeting LNM-negative CRC to guide immunotherapy decisions. To reduce heterogeneity, our analysis was limited to LNM-negative cases; however, immune profiles may differ in LNM-positive patients. Further studies are needed to assess the generalizability of these subtypes across CRC stages.

Activation of the MAPK pathway through *KRAS* influences cell proliferation and differentiation, potentially preventing tumorigenesis [25]. CRCs with BRAF mutations is associated with lymphocyte infiltration and immune response activation [26], with studies indicating a positive correlation between PD-L1 expression and BRAF mutations. High CD8＋tumor-infiltrating lymphocytes observed in BRAF-mutant CRC suggest that these patients may respond well to immunotherapy [27]. Although *KRAS* mutations are generally unrelated to LNM status or tumor characteristics [4,9], our study demonstrates a shift in *KRAS* mutation rates across the two LNM-negative subtypes, potentially offering survival benefits to CRC patients with BRAF mutations.

To identify subtype-specific diagnostic biomarkers, we analyzed differentially expressed genes between the C1 and C2 subtypes. As shown in S2 Table, *PPFIA4* was among the most significantly upregulated genes in subtype C1. *PPFIA4* encodes the protein liprin-α4, which has been implicated in promoting tumor glycolysis and angiogenesis via Wnt signaling [22], and may contribute to colorectal cancer progression through immune modulation [21]. Our in vitro experiments demonstrated that *PPFIA4* knockdown significantly inhibited CRC cell proliferation and migration, supporting its functional relevance in tumor progression. These findings suggest that *PPFIA4* is not only a marker of immune subtype C1, but may also represent a potential therapeutic target in LNM-negative CRC. Beyond its biological function, *PPFIA4* holds promise as a clinically actionable biomarker, potentially serving both diagnostic and therapeutic purposes. Its subtype-specific expression profile could help distinguish more aggressive immune subtypes in early-stage CRC, and its inhibition may offer a new target for anti-tumor intervention. However, as our findings are based solely on in vitro data, further validation through in vivo studies using patient tissues or animal models is necessary to confirm its clinical utility.

In conclusion, we conducted an in-depth analysis of the immune signature of LNM-negative CRC, especially regarding the relationship between immune subtypes and immunotherapy response. These findings enhance our understanding of CRC progression and may inform personalized strategies for cancer immunotherapy.

## Supporting information

**S1 Table. The 109 immune associated genes used for classification.**
(XLSX)

**S2 Table. The result of differential expression analysis.**
(XLSX)

**S3 Table. Functional enrichment analyses of subclass specific genes.**
(XLSX)

**S4 Table. Pathway enrichment analysis of two CRC subclasses.**
(XLSX)

**S5 Table. P values for gene set mRNA enrichment analysis.**
(XLSX)

**S6 Table. The abundances of 13 immune-related cells.**
(XLSX)

**S7 Table. Expression profiles of 19 potential targeted immune checkpoint genes in two subtypes.**
(XLSX)

**S8 Table. TIDE value of immune checkpoint suppression therapy for each sample (Sheet 8).**
(XLSX)

**S9 Table. Genetic mutation analysis of GSE39582 dataset.**
(XLSX)

**S10 Table. ICI scores for each sample in TCGA and GSE39582.**
(XLSX)

**S1 Fig. Validation of molecular subclasses in LNM-negative CRC using the GSE103479 dataset.** (A) Use immune-related genes to perform NMF clustering on the GSE103479. Cophenetic correlation coefficient for k = 2–6 is shown. (B) Overall survival (OS) analysis of the two subclasses (C1 and C2) in GSE103479, with statistical significance assessed by log-rank test.
(PDF)

**S1 File. Raw Western blot images (Raw_Images_All_.pdf).** Original, uncropped Western blot images corresponding to the results shown in Figure 6B (liprin a4 and GAPDH in SW480 and HCT-116 cells).
(PDF)

## Author contributions

**Conceptualization:** Wenliang Yuan.

**Data curation:** Wenliang Yuan.

**Formal analysis:** Li Liu.

**Funding acquisition:** Wenliang Yuan, Li Liu.

**Investigation:** Li Liu.

**Methodology:** Wenliang Yuan.

**Resources:** Li Liu.

**Validation:** Wenliang Yuan.

**Visualization:** Li Liu.

**Writing – original draft:** Wenliang Yuan.

**Writing – review & editing:** Li Liu.

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
