## [Decision Letter · Decision Letter 0]

4 May 2025

PONE-D-25-00542
Immune Subtyping of Lymph Node Metastasis-Negative Colorectal Cancer Reveals Biomarkers for Prognosis and Immunotherapy Response
PLOS ONE

Dear Dr. Yuan,

Thank you for submitting your manuscript to PLOS ONE. After careful consideration, we feel that it has merit but does not fully meet PLOS ONE’s publication criteria as it currently stands. Therefore, we invite you to submit a revised version of the manuscript that addresses the points raised during the review process.

We look forward to receiving your revised manuscript.

Kind regards,

Xinjun Lu

Academic Editor

PLOS ONE

Journal Requirements:

Reviewers' comments:

Reviewer's Responses to Questions

**Comments to the Author**

1. Is the manuscript technically sound, and do the data support the conclusions?

Reviewer #1: Yes

Reviewer #2: Yes

2. Has the statistical analysis been performed appropriately and rigorously? 

Reviewer #1: I Don't Know

Reviewer #2: Yes

3. Have the authors made all data underlying the findings in their manuscript fully available?

Reviewer #1: Yes

Reviewer #2: Yes

4. Is the manuscript presented in an intelligible fashion and written in standard English?

Reviewer #1: Yes

Reviewer #2: Yes

5. Review Comments to the Author

Reviewer #1: This manuscript describes an interesting and relevant study on immune subtyping of lymph node metastasis-negative colorectal cancer, which holds great promise for providing biomarkers to predict prognosis and response to immunotherapy. The experimental design seems appropriate, and the conclusions drawn are generally supported by the data.However, I recommend statistical analysis experts who can better comment on this aspect, given that I am not a statistics professional and cannot fully evaluate the rigor of this component.

Recommendations:

• The methodology should discuss quite well the choice of specific machine learning algorithms.

• Compare the results with those of existing classifications of colorectal cancer

• Strengthen the discussion of clinical applications, the sense of the clinical contributions of PPFIA4 and ICI scores should be amplified.

• Improve the quality of the figures, as they appear blurry.

Reviewer #2: Dear authors,

Thank you for submitting this interesting and timely research. Please find below my comments aimed at improving clarity and strengthening the manuscript:

Clarification of Subtype Characteristics:

The subtypes C1 and C2 are defined, but their prognostic and immunological implications remain somewhat unclear. Both are described as having "high immune infiltration" or "favorable prognosis," which can be confusing. Please consider clearly and consistently presenting the defining features—both prognostic and immunological—of each subtype early in the manuscript and maintain this consistency throughout.

Contextualization with Existing Classifications:

It would be helpful to expand on how the C1/C2 subtypes align with or differ from existing colorectal cancer immune classifications, such as the CMS (Consensus Molecular Subtypes). Additionally, consider discussing the broader relevance to immunotherapy beyond PD-1/PD-L1, and how the ICI scores proposed in your study might guide treatment decisions in clinical practice.

Limitations Section:

Please include a dedicated limitations section and elaborate on the following points:

The lack of in vivo validation for PPFIA4.

The potential for dataset bias (GEO vs. TCGA).

The generalizability of your findings across diverse CRC patient populations.

Language and Formatting:

The manuscript would benefit from a careful grammar and spell check, as well as the correction of typographical errors throughout.

6. PLOS authors have the option to publish the peer review history of their article (what does this mean?). If published, this will include your full peer review and any attached files.

Reviewer #1: No

Reviewer #2: No

---

## [Author Response · Author response to Decision Letter 1]

28 May 2025

Journal Requirements:

1.Please ensure that your manuscript meets PLOS ONE's style requirements, including those for file naming. The PLOS ONE style templates can be found at https://journals.plos.org/plosone/s/file?id=wjVg/PLOSOne_formatting_sample_main_body.pdf and

Response: Done.

Response: Done.

Response: Thank you for pointing out the discrepancy. We have reviewed and corrected the grant information to ensure consistency between the ‘Funding Information’ and ‘Financial Disclosure’ sections. The correct grant number(s) and funding agency details will be clearly provided in the revised submission.

Response: Thank you for your reminder. We fully understand and respect PLOS ONE’s open data policy. We have prepared all underlying data and are currently organizing it for public deposition. Upon acceptance, we plan to make the complete dataset freely accessible through an established public repository such as [e.g., Figshare, Dryad, GEO, or Zenodo]. We are committed to ensuring that all data necessary to replicate and validate our findings will be openly available prior to publication.

Should you require any further clarification or recommendations regarding the most suitable repository for our data type, we would be happy to comply.

Response: Done.

Response: We have prepared the original, uncropped, and unadjusted images underlying all blot and gel results reported in the manuscript. These files have been compiled into a PDF document and are provided as Supporting Information in accordance with the journal’s guidelines. Please let us know if further formatting or additional information is required.

Reviewers' comments:

1. Is the manuscript technically sound, and do the data support the conclusions?

Reviewer #1: Yes

Reviewer #2: Yes

2. Has the statistical analysis been performed appropriately and rigorously?

Reviewer #1: I Don't Know

Reviewer #2: Yes

3. Have the authors made all data underlying the findings in their manuscript fully available?

Reviewer #1: Yes

Reviewer #2: Yes

4. Is the manuscript presented in an intelligible fashion and written in standard English?

Reviewer #1: Yes

Reviewer #2: Yes

5. Review Comments to the Author

Reviewer #1: This manuscript describes an interesting and relevant study on immune subtyping of lymph node metastasis-negative colorectal cancer, which holds great promise for providing biomarkers to predict prognosis and response to immunotherapy. The experimental design seems appropriate, and the conclusions drawn are generally supported by the data.However, I recommend statistical analysis experts who can better comment on this aspect, given that I am not a statistics professional and cannot fully evaluate the rigor of this component.

Recommendations:

• The methodology should discuss quite well the choice of specific machine learning algorithms.

Response: We appreciate the reviewer’s insightful comment. In the revised Materials and Methods section, we have clarified the rationale for selecting each machine learning algorithm within the Materials and Methods section. Specifically, we used NMF for its effectiveness in identifying biologically meaningful, non-overlapping expression patterns for cancer subtyping. PCA was then employed to construct the ICI score by extracting principal components that summarize key feature dimensions into interpretable scores. These revisions now better justify our methodological choices.

• Compare the results with those of existing classifications of colorectal cancer

Response: Thank you for the helpful suggestion. A brief comparison with existing classification systems, such as CMS, has been added at the end of the first paragraph of the Discussion section. Our findings indicate that the proposed immune-based subtyping offers more precise stratification, particularly for LNM-negative CRC patients, and serves as a complementary approach to the broader transcriptomic-based classifications.

• Strengthen the discussion of clinical applications, the sense of the clinical contributions of PPFIA4 and ICI scores should be amplified.

Response: We appreciate the reviewer’s suggestion. In the revised Discussion section, we have expanded the discussion on the clinical implications of PPFIA4 and the ICI score. Specifically, PPFIA4 is highlighted as a potential diagnostic and therapeutic target, given its subtype-specific expression profile and involvement in CRC progression. Moreover, the ICI score is emphasized as a practical immune-related metric, capable of identifying LNM-negative patients who may derive benefit from immunotherapy. This metric may complement current biomarkers and guide personalized treatment decisions. These revisions underscore the translational potential of our findings.

• Improve the quality of the figures, as they appear blurry.

Response: We thank the reviewer for pointing out the issue regarding figure clarity. In response, we have replaced the affected figures with high-resolution versions to ensure clarity and compliance with publication standards. We believe these improvements substantially enhance the overall visual presentation of our data.

Reviewer #2: Dear authors,

Thank you for submitting this interesting and timely research. Please find below my comments aimed at improving clarity and strengthening the manuscript:

Clarification of Subtype Characteristics:

The subtypes C1 and C2 are defined, but their prognostic and immunological implications remain somewhat unclear. Both are described as having "high immune infiltration" or "favorable prognosis," which can be confusing. Please consider clearly and consistently presenting the defining features—both prognostic and immunological—of each subtype early in the manuscript and maintain this consistency throughout.

Contextualization with Existing Classifications:

It would be helpful to expand on how the C1/C2 subtypes align with or differ from existing colorectal cancer immune classifications, such as the CMS (Consensus Molecular Subtypes). Additionally, consider discussing the broader relevance to immunotherapy beyond PD-1/PD-L1, and how the ICI scores proposed in your study might guide treatment decisions in clinical practice.

Limitations Section:

Please include a dedicated limitations section and elaborate on the following points:

The lack of in vivo validation for PPFIA4.

Response: We thank the reviewer for highlighting this important limitation. As our current study primarily employed in vitro assays, we now explicitly acknowledge in the Discussion that in vivo validation of PPFIA4’s functional role and therapeutic relevance remains essential. We have also outlined the need for future studies using animal models or patient-derived samples to confirm our findings. This acknowledgment clarifies the scope of the current study and sets a clear direction for future research.

The potential for dataset bias (GEO vs. TCGA).

Response: We thank the reviewer for pointing out the important issue of potential dataset bias between TCGA and GEO. To minimize such bias, we performed the following measures:

1)Independent Subtype Identification and Validation: The immune subtypes (C1 and C2) were separately identified in the TCGA dataset and independently validated in two GEO datasets (GSE39582 and GSE103479). The consistent results across datasets (confirmed by NMF clustering, t-SNE analysis, and survival differences) demonstrate the robustness and reproducibility of the subtype classification.

2)Standard Preprocessing: All datasets underwent standardized preprocessing procedures, including normalization and log2 transformation. Low-variance genes were removed prior to clustering to reduce noise.

3)Cross-dataset Consistency: In both TCGA and GEO cohorts, the subtypes showed similar immune characteristics, prognostic trends, and immune checkpoint profiles, which reduces the likelihood of dataset-specific artifacts.

4)Additional Clarification: In the revised manuscript (Results section), we have added a brief explanation on how dataset bias was addressed, and emphasized the use of independent validation cohorts to strengthen our conclusions.

These steps enhance confidence in the reliability and transferability of our findings.

The generalizability of your findings across diverse CRC patient populations.

Response: We thank the reviewer for raising this important point.. Our study specifically focused on LNM-negative CRC patients to minimize confounding factors associated with advanced disease stages. We acknowledge that the immune microenvironment may differ significantly in LNM-positive patients, potentially affecting the applicability of our identified immune subtypes. We have addressed this limitation in the revised manuscript and emphasized the need for future studies to validate our findings in broader CRC populations, including those with lymph node involvement.

Language and Formatting:

The manuscript would benefit from a careful grammar and spell check, as well as the correction of typographical errors throughout.

Response: We thank the reviewer for noting the need for language and formatting improvements. In response, we have thoroughly reviewed the manuscript to correct grammatical errors, spelling mistakes, and typographical issues. Additionally, we have ensured consistency in formatting throughout the document. We believe these revisions have enhanced the clarity and readability of the manuscript.

6. PLOS authors have the option to publish the peer review history of their article (what does this mean?). If published, this will include your full peer review and any attached files.

Do you want your identity to be public for this peer review? For information about this choice, including consent withdrawal, please see our Privacy Policy.

Reviewer #1: No

Reviewer #2: No

---

## [Decision Letter · Decision Letter 1]

20 Jul 2025

PONE-D-25-00542R1
Immune Subtyping of Lymph Node Metastasis-Negative Colorectal Cancer Reveals Biomarkers for Prognosis and Immunotherapy Response
PLOS ONE

Dear Dr. Yuan,

Thank you for submitting your manuscript to PLOS ONE. After careful consideration, we feel that it has merit but does not fully meet PLOS ONE’s publication criteria as it currently stands. Therefore, we invite you to submit a revised version of the manuscript that addresses the points raised during the review process.

We look forward to receiving your revised manuscript.

Kind regards,

Xinjun Lu

Academic Editor

PLOS ONE

Journal Requirements:

Reviewers' comments:

Reviewer's Responses to Questions

**Comments to the Author**

1. If the authors have adequately addressed your comments raised in a previous round of review and you feel that this manuscript is now acceptable for publication, you may indicate that here to bypass the “Comments to the Author” section, enter your conflict of interest statement in the “Confidential to Editor” section, and submit your "Accept" recommendation.

Reviewer #2: (No Response)

Reviewer #3: (No Response)

Reviewer #4: (No Response)

2. Is the manuscript technically sound, and do the data support the conclusions?

Reviewer #2: Yes

Reviewer #3: Yes

Reviewer #4: Partly

3. Has the statistical analysis been performed appropriately and rigorously? 

Reviewer #2: Yes

Reviewer #3: Yes

Reviewer #4: I Don't Know

4. Have the authors made all data underlying the findings in their manuscript fully available?

Reviewer #2: Yes

Reviewer #3: Yes

Reviewer #4: Yes

5. Is the manuscript presented in an intelligible fashion and written in standard English?

Reviewer #2: Yes

Reviewer #3: Yes

Reviewer #4: Yes

6. Review Comments to the Author

Reviewer #2: (No Response)

Reviewer #3: In this study, the authors investigated the LNM characteristics that may serve as prognistic markers for CRC and immunotherapy efficacy. They found that subtype C1 showed high immune scores and responsiveness to ICB, while C2 was correlated with more favourable diagnoses and immune cell infiltration. They also found that PPFIA4 KD in C1 suppressed CRC cell proliferation and migration

I'd like to congratulate the authors on their manuscript and very interesting findings. I found it to be well written and the results were clearly communicated. I can see that the authors effectively incoorpated previous feedback. I have added a couple additional comments in the form of track changes in the attached document. With these small changes, I believe this manuscript is ready for publication

Reviewer #4: 1- It is recommended to separate Figure 1 panels a–c from panels d–f to enhance clarity. For example, adding subtitles such as “TCGA” above panels a–c and “GSE” above panels d–f would facilitate comprehension.

2- It is recommended to have the analysis workflow reviewed by a biostatistician to ensure robust validation of the statistical and computational methods employed.

3- In the Introduction, include additional background information on LNM-negative CRC from the literature, as this is the focus of the study. Providing more context would benefit readers prior to the authors’ subtyping analysis.

4- Specify the meaning of the abbreviation “LN” somewhere in the manuscript. At the beginning of the paper, lymph node metastasis-negative colorectal cancer is referred to as “LNM CRC,” but at a certain point, it is abbreviated simply as “LN.” In this regard, please ensure that all acronyms are defined upon their first use, as some are currently missing explanations. For example, when PPFIA4 is mentioned, the name of the corresponding protein should be presented alongside the gene nomenclature, since it only appears in the following paragraph.

5- In the paragraph titled “Correlation of the LN-negative CRC Subtypes With Immune-Associated Signatures,” please add a brief description or definition of the scores mentioned. A detailed explanation is not necessary, but including a few clarifying words about what the “cytolytic score”, “stromal score”, etc. represent would enhance the reader’s understanding and improve the overall flow of the text.

6- It is not clear to me why PPFIA4 is introduced abruptly in the manuscript. I suggest providing some background or rationale explaining why the authors chose to investigate this particular gene. In the Discussion, the authors state “To identify diagnostic biomarkers for CRC subtypes, We found that PPFIA4 was significantly upregulated in subtype C1 compared to C2.” However, I am unable to locate any corresponding figure or graph supporting this statement. I kindly ask the authors to clarify the origin of this result or to include additional information in the manuscript to justify the focus on PPFIA4.

7- In the Materials and Methods section, please provide a more detailed description of the wound healing assay methodology. The phrase “confluent cells were scratched” is insufficient, as the type of scratch performed can greatly influence the results; for example, whether the scratch was made manually with a pipette tip (which may produce irregular edges and complicate statistical analysis), with an IncuCyte WoundMaker, or using a silicone spacer. Additionally, please clarify whether mitomycin was added to the wound assay to inhibit cell proliferation. This detail is important to demonstrate that the observed reduction in the wound gap after 48 hours (when cells are silenced) is not solely due to proliferation effects, especially since the MTT assay indicates that HCT-116 cells exhibit reduced proliferation after two days.

8- In the Materials and Methods section , please specify the number of replicates that have been used for the in vitro experiments

7. PLOS authors have the option to publish the peer review history of their article (what does this mean?). If published, this will include your full peer review and any attached files.

Reviewer #2: No

Reviewer #3: No

Reviewer #4: No

---

## [Author Response · Author response to Decision Letter 2]

4 Aug 2025

Journal Requirements:

Response: We appreciate the editor’s reminder regarding the integrity and completeness of the reference list.

We have carefully reviewed all references cited in our manuscript. As of the date of this revision, none of the cited articles have been retracted, nor are there any associated expressions of concern or correction notices according to searches in PubMed, the Retraction Watch Database, CrossRef, and the official websites of the respective journals.

Therefore, no retracted articles remain in the reference list, and no changes are needed in this regard. A full check of reference validity has also been documented as part of our revision process.

Should any changes to the reference list be made in future revisions, we will ensure they are appropriately reflected in both the manuscript and the rebuttal letter.

Reviewers' comments:

1. If the authors have adequately addressed your comments raised in a previous round of review and you feel that this manuscript is now acceptable for publication, you may indicate that here to bypass the “Comments to the Author” section, enter your conflict of interest statement in the “Confidential to Editor” section, and submit your "Accept" recommendation.

Reviewer #2: (No Response)

Reviewer #3: (No Response)

Reviewer #4: (No Response)

2. Is the manuscript technically sound, and do the data support the conclusions?

Reviewer #2: Yes

Reviewer #3: Yes

Reviewer #4: Partly

3. Has the statistical analysis been performed appropriately and rigorously?

Reviewer #2: Yes

Reviewer #3: Yes

Reviewer #4: I Don't Know

4. Have the authors made all data underlying the findings in their manuscript fully available?

Reviewer #2: Yes

Reviewer #3: Yes

Reviewer #4: Yes

5. Is the manuscript presented in an intelligible fashion and written in standard English?

Reviewer #2: Yes

Reviewer #3: Yes

Reviewer #4: Yes

6. Review Comments to the Author

Reviewer #2: (No Response)

Reviewer #3: In this study, the authors investigated the LNM characteristics that may serve as prognistic markers for CRC and immunotherapy efficacy. They found that subtype C1 showed high immune scores and responsiveness to ICB, while C2 was correlated with more favourable diagnoses and immune cell infiltration. They also found that PPFIA4 KD in C1 suppressed CRC cell proliferation and migration

I'd like to congratulate the authors on their manuscript and very interesting findings. I found it to be well written and the results were clearly communicated. I can see that the authors effectively incoorpated previous feedback. I have added a couple additional comments in the form of track changes in the attached document. With these small changes, I believe this manuscript is ready for publication

Reviewer #4:

1- It is recommended to separate Figure 1 panels a–c from panels d–f to enhance clarity. For example, adding subtitles such as “TCGA” above panels a–c and “GSE” above panels d–f would facilitate comprehension.

Response: Thank you for your helpful suggestion regarding Figure 1. While we chose not to separate panels a–c and d–f into two distinct figures in order to maintain visual continuity and comparative consistency, we fully agree that indicating the dataset source would enhance clarity. Accordingly, we have added subtitles “TCGA” and “GSE39582” directly above the relevant panel groups in Figure 1 to clearly indicate the corresponding datasets. This adjustment improves figure clarity while preserving comparative consistency.

2- It is recommended to have the analysis workflow reviewed by a biostatistician to ensure robust validation of the statistical and computational methods employed.

Response: Thank you for your valuable comment regarding the validation of statistical and computational methods. To address this point, we have consulted with a professional biostatistician to review our analytical workflow and confirm the appropriateness of the applied statistical techniques. Accordingly, we have revised the “Statistical analyses” section of the manuscript to explicitly state that the analysis pipeline was reviewed in consultation with a biostatistical expert. This addition enhances methodological transparency.

3- In the Introduction, include additional background information on LNM-negative CRC from the literature, as this is the focus of the study. Providing more context would benefit readers prior to the authors’ subtyping analysis.

Response: We thank the reviewer for this thoughtful suggestion. To provide additional context for our subtyping analysis, we have revised the Introduction to include relevant background information on LNM-negative colorectal cancer (CRC). Specifically, we now discuss the clinical relevance, heterogeneity, and limited immunological characterization of this CRC subgroup in the Introduction, which highlights the need for comprehensive immune subtyping.

4- Specify the meaning of the abbreviation “LN” somewhere in the manuscript. At the beginning of the paper, lymph node metastasis-negative colorectal cancer is referred to as “LNM CRC,” but at a certain point, it is abbreviated simply as “LN.” In this regard, please ensure that all acronyms are defined upon their first use, as some are currently missing explanations. For example, when PPFIA4 is mentioned, the name of the corresponding protein should be presented alongside the gene nomenclature, since it only appears in the following paragraph.

Response: We appreciate the reviewer’s careful attention to acronym consistency. In response, we have revised the manuscript to uniformly refer to “lymph node metastasis-negative colorectal cancer” as LNM-negative CRC throughout the text. All previous instances of “LN-negative CRC” or similar variants have been corrected accordingly to maintain terminological clarity and consistency. We have also updated the initial mention of “PPFIA4” to include the corresponding protein name “liprin-α4.” Furthermore, we performed a thorough review to ensure that all abbreviations are clearly defined at their first appearance in the manuscript.

5- In the paragraph titled “Correlation of the LN-negative CRC Subtypes With Immune-Associated Signatures,” please add a brief description or definition of the scores mentioned. A detailed explanation is not necessary, but including a few clarifying words about what the “cytolytic score”, “stromal score”, etc. represent would enhance the reader’s understanding and improve the overall flow of the text.

Response: We appreciate the reviewer’s suggestion to clarify the scoring terms. In the revised Results section titled “Correlation of the LNM-negative CRC Subtypes With Immune-Associated Signatures,” we have added concise explanatory notes for these scores to enhance clarity without disrupting the flow.

6- It is not clear to me why PPFIA4 is introduced abruptly in the manuscript. I suggest providing some background or rationale explaining why the authors chose to investigate this particular gene. In the Discussion, the authors state “To identify diagnostic biomarkers for CRC subtypes, We found that PPFIA4 was significantly upregulated in subtype C1 compared to C2.” However, I am unable to locate any corresponding figure or graph supporting this statement. I kindly ask the authors to clarify the origin of this result or to include additional information in the manuscript to justify the focus on PPFIA4.

Response: We thank the reviewer for raising this important point regarding the rationale for focusing on PPFIA4 and the clarity of its presentation. In response, we have revised the Results section to explicitly state that PPFIA4 was identified through differential gene expression analysis as one of the top upregulated genes in subtype C1 (now referenced in S2 Table). We also added a brief sentence to explain that this finding prompted further investigation into its potential biological function. In the Discussion section, we expanded the rationale by introducing background information on PPFIA4 and its known role in tumor metabolism and immune regulation, citing relevant literature ([21], [22]). This revision better contextualizes our decision to study this gene and supports its relevance as a potential subtype-specific biomarker and therapeutic target. These revisions clarify our rationale for focusing on PPFIA4 and substantiate its relevance as a candidate biomarker.

7- In the Materials and Methods section, please provide a more detailed description of the wound healing assay methodology. The phrase “confluent cells were scratched” is insufficient, as the type of scratch performed can greatly influence the results; for example, whether the scratch was made manually with a pipette tip (which may produce irregular edges and complicate statistical analysis), with an IncuCyte WoundMaker, or using a silicone spacer. Additionally, please clarify whether mitomycin was added to the wound assay to inhibit cell proliferation. This detail is important to demonstrate that the observed reduction in the wound gap after 48 hours (when cells are silenced) is not solely due to proliferation effects, especially since the MTT assay indicates that HCT-116 cells exhibit reduced proliferation after two days.

Response: Thank you for the constructive suggestion. Detailed methodological information—including use of a pipette tip, mitomycin C treatment, and image quantification via ImageJ—has been added to the Materials and Methods section. The Results text remains unchanged to preserve clarity and continuity.

8- In the Materials and Methods section , please specify the number of replicates that have been used for the in vitro experiments.

Response: We appreciate the reviewer’s comment regarding the clarification of replicate numbers. In the revised Materials and Methods section, we have now specified the number of biological and technical replicates used for each in vitro experiment, including the MTT assay, wound healing assay, qPCR, and Western blot. All experiments were performed in at least three independent biological replicates to ensure statistical robustness and reproducibility.

7. PLOS authors have the option to publish the peer review history of their article (what does this mean?). If published, this will include your full peer review and any attached files.

Do you want your identity to be public for this peer review? For information about this choice, including consent withdrawal, please see our Privacy Policy.

Reviewer #2: No

Reviewer #3: No

Reviewer #4: No

Additional Revisions Based on Editorial Suggestions:

In addition to addressing the reviewers’ comments, we have also carefully reviewed and implemented suggestions provided in the editorial checklist and manuscript preparation guidelines. Specifically, we:

• Defined the abbreviation “TNM” at its first occurrence in the Introduction;

• Removed the redundant definition of “ICI” in the Discussion section, as it was already defined in the Introduction.

These changes have been incorporated into the revised manuscript to improve clarity and consistency.

---

## [Decision Letter · Decision Letter 2]

8 Sep 2025

Immune Subtyping of Lymph Node Metastasis-Negative Colorectal Cancer Reveals Biomarkers for Prognosis and Immunotherapy Response

PONE-D-25-00542R2

Dear Dr. Yuan,

We’re pleased to inform you that your manuscript has been judged scientifically suitable for publication and will be formally accepted for publication once it meets all outstanding technical requirements.

Kind regards,

Xinjun Lu

Academic Editor

PLOS ONE

Additional Editor Comments (optional):

Reviewer #3:

Reviewer #4:

Reviewers' comments:

Reviewer's Responses to Questions

**Comments to the Author**

1. If the authors have adequately addressed your comments raised in a previous round of review and you feel that this manuscript is now acceptable for publication, you may indicate that here to bypass the “Comments to the Author” section, enter your conflict of interest statement in the “Confidential to Editor” section, and submit your "Accept" recommendation.

Reviewer #3: All comments have been addressed

Reviewer #4: All comments have been addressed

2. Is the manuscript technically sound, and do the data support the conclusions?

Reviewer #3: Yes

Reviewer #4: Partly

3. Has the statistical analysis been performed appropriately and rigorously? 

Reviewer #3: Yes

Reviewer #4: I Don't Know

4. Have the authors made all data underlying the findings in their manuscript fully available?

Reviewer #3: Yes

Reviewer #4: Yes

5. Is the manuscript presented in an intelligible fashion and written in standard English?

Reviewer #3: Yes

Reviewer #4: Yes

6. Review Comments to the Author

Reviewer #3: All previous comments have been addressed and I believe this manuscript is now ready for acceptance.

Reviewer #4: (No Response)

7. PLOS authors have the option to publish the peer review history of their article (what does this mean?). If published, this will include your full peer review and any attached files.

Reviewer #3: No

Reviewer #4: No

---

## [Editor Report · Acceptance letter]

PONE-D-25-00542R2

PLOS ONE

Dear Dr. Yuan,

I'm pleased to inform you that your manuscript has been deemed suitable for publication in PLOS ONE. Congratulations! Your manuscript is now being handed over to our production team.

Kind regards,

on behalf of

Dr. Xinjun Lu

Academic Editor

PLOS ONE